# Nutritional Status and Diet Style Affect Cognitive Function in Alcoholic Liver Disease

**DOI:** 10.3390/nu13010185

**Published:** 2021-01-09

**Authors:** Ye Rin Choi, Hyeong Seop Kim, Sang Jun Yoon, Na Young Lee, Haripriya Gupta, Ganesan Raja, Yoseph Asmelash Gebru, Gi Soo Youn, Dong Joon Kim, Young Lim Ham, Ki Tae Suk

**Affiliations:** 1Institute for Liver and Digestive Diseases, Hallym University College of Medicine, Chuncheon 24252, Korea; dpfls3020@gmail.com (Y.R.C.); kimhs2425@gmail.com (H.S.K.); ysjtlhuman@gmail.com (S.J.Y.); na0lee0218@gmail.com (N.Y.L.); phr.haripriya13@gmail.com (H.G.); vraja.ganesan@gmail.com (G.R.); yagebru@gmail.com (Y.A.G.); gisu0428@hallym.ac.kr (G.S.Y.); djkim@hallym.ac.kr (D.J.K.); 2Department of Nursing, Daewon University College, Jaecheon 27135, Korea

**Keywords:** alcoholic liver disease, cognitive function, calorie intake, nutrition, BMI

## Abstract

Malnutrition and cognitive dysfunction are typical features of alcoholic liver disease (ALD) and are correlated with the development of complications. The aim of this study is to explore the effect of nutritional state and diet on cognitive function in ALD. A total of 43 patients with compensated alcoholic cirrhosis were enrolled, and a neuropsychological test was assessed according to body mass index (BMI, <22 and ≥22). In the ALD animal study, mice were divided into five groups (*n* = 9/group; normal liquid, 5% EtOH + regular liquid, 5% EtOH + high-carbohydrate liquid, 5% EtOH + high-fat liquid, and 5% EtOH + high-protein liquid diet) and fed the same calories for eight weeks. To assess cognitive function, we performed T-maze studies weekly before/after alcohol binging. In cognitive function (BMI < 22/≥22), language score of Korea mini-mental state (7.4 ± 1.4/7.9 ± 0.4), Boston naming (11.7 ± 2.7/13.0 ± 1.8), forward digit span (6.7 ± 1.8/7.5 ± 1.6), Korean color word stroop (24.2 ± 26.5/43.6 ± 32.4), and interference score (33.9 ± 31.9/52.3 ± 33.9) revealed significant differences. In the T-maze test, alcohol significantly delayed the time to reach food, and binge drinking provided a temporary recovery in cognition. The alcohol-induced delay was significantly reduced in the high-carbohydrate and high-fat diet groups. Synaptic function exhibited no changes in all groups. Cognitive dysfunction is affected by nutritional status and diet in ALD.

## 1. Introduction

Alcoholic liver disease (ALD) is the third most common cause of chronic liver disease and shows high mortality rate worldwide [1]. The number of alcohol-related mortality reaches about at 2.5 million annually, occupying 4% of all deaths worldwide [2]. Excessive and binge alcohol drinking reveal many adverse medical and social results [3,4]. Alcohol abuse among social subjects is common and occurs more frequently. Moreover, it is associated with cognitive impairment and independently with short- and long-term mortality [5].

Most patients with ALD consume a deficient diet and exhibit malnutrition [6]. Malnutrition is one of the major complications of ALD that has been studied recently, especially in patients with severe alcoholic hepatitis [7]. Malnutrition worsens clinical outcome in ALD, and nutritional support improves nutritional status and may improve clinical outcome. Possible reasons for malnutrition include metabolic disturbance, nausea, vomiting, or unbalanced diet [8].

Alcohol consumption can cause functional damage of brain and change of frontal lobe because alcohol has a toxic and chemical effects [9,10,11]. Neuropsychological dysfunction due to frontal lobe abnormalities is similar to that noted in chronic alcohol abuse. Consequently, we can conclude that chronic alcohol drinking and chronic liver disease are independently correlated with changes in neuropsychological dysfunction. In this regard, maintaining an adequate nutritional status is essential for ALD patients.

Many studies show that nutrition status is positively related with cognitive function. A high-calorie diet reduces hippocampal synaptic plasticity and impairs cognitive function through brain-derived neurotrophic factor-mediated effects on dendritic spines [12]. Furthermore, dietary fat intake at midlife affects cognitive performance. Based on this evidence, a patient’s nutritional state is essential to prevent complications, and important lifestyle factor that can alter the risk of cognitive impairment in the long-term [13,14]. 

In our previous report, impaired memory and frontal lobe executive functions and early development of overt encephalopathy were more common in patients with ALD [9]. However, limited data about the relation between malnutrition and cognitive dysfunction in ALD are available. The aim of this study is to explore the effect of nutrition on cognitive function in ALD. 

## 2. Materials and Methods

### 2.1. Patients

A total of 43 patients with alcoholic liver cirrhosis were prospectively recruited from October 2011 to March 2020 (NCT04557774). The diagnosis of liver disease was performed based on laboratory data, endoscopic findings, medical record review, and liver biopsy. Patients who were >20 years old with liver function test results indicating an aspartate aminotransferase (AST)/alanine aminotransferase (ALT) > 1 and elevated AST (ALT) levels as well as a history of alcohol consumption of greater than 40 g/day for women and 60 g/day for men during the 7 days before screening were enrolled. Their last drinks were consumed within 48 h prior to admission. Patients with a history of hepatitis virus infection and those who were administered medications for sedation, seizure, head trauma, stroke, dementia, Parkinson’s disease, or any kind of focal neurologic deficits were excluded. Any patients who were suspected of alcohol-induced direct neurologic damage, such as Wernicke’s encephalopathy, alcohol-induced spinal cord disease, or alcohol-induced peripheral nerve disease, were excluded. 

Patients were recruited after receiving the approval of institutional review board of Hallym University Chuncheon Sacred Heart Hospital (2011-36). Caloric intake was considered to affect cognitive function and was arbitrarily divided according to the nutritional status (BMI < 22 and BMI ≥ 22) of 43 ALD patients.

We conducted baseline evaluations, including family and alcohol history, X-ray, electrocardiography, blood tests for electrolytes, liver function, and viral markers. Serum biochemical parameters included total bilirubin, AST, ALT, gamma-glutamyltranspeptidase (γ-GT), alkaline phosphatase (ALP), albumin, blood urea nitrogen, creatinine, α-fetoprotein, prothrombin time, blood glucose, triglycerides, and total cholesterol. Child-Pugh and MELD scores were evaluated based on laboratory and imaging findings. 

To select patients with the same medical condition, endoscopy and imaging (abdominal computed tomography (CT) or ultrasound) were performed for all the patients. The hepatic venous pressure gradient was determined by subtracting free hepatic venous pressure from wedged hepatic venous pressure. In addition, a brain CT was performed on patients who consented.

The diagnosis of liver cirrhosis was established by liver biopsy and/or imaging studies, such as ultrasound and/or contrast-enhanced CT, in conjunction with laboratory data and clinical complications of cirrhosis [15]. 

### 2.2. Neuropsychological Test

In this study, we performed neuropsychological tests in patients with alcoholic liver cirrhosis to evaluate the effect of BMI on cognitive function. We commissioned the cognitive function test to the neurological function laboratory of Chuncheon Sacred Heart Hospital. We used the Seoul Neuropsychological Screening Battery (SNSB, Human Brain Research & Consulting Co, Seoul, Korea) as the neuropsychological test. Attention function (digit-span forward/backward and letter cancellation), frontal/executive function (Controlled oral ward association test (COWAT), Korean color word stroop test (K-CWST)), visuospatial function (Korea-mini mental status examination (K-MMSE) and Ray-complex figure test (RCFT) copy score), verbal memory (Seoul-verbal learning test (SVLT)), and visual memory (RCFT delayed recall) were assessed in all enrolled subjects. The results were compared using absolute scores instead of percentile scores. Test results, including monitoring of scoring, procession and interpretation, were evaluated by one neurology specialist.

### 2.3. Experimental Animals

After clinical study analyses, we conducted alcohol animal experiments in mice fed a high-protein, high-carbohydrate, and high-fat diet purchased from Dooyeol Biotech (Seoul, Korea). Six-week-old male C57Bl/6J mice were obtained for the alcohol and diet experiment. We used 20 mice to evaluate changes of cognitive function in alcoholic liver disease. To induce the effect of diet in alcohol mice model, 45 mice were subdivided into 5 groups (*n* = 9/group; normal regular liquid diet, control (5% EtOH liquid diet + regular liquid diet], high carbohydrate (5% EtOH + high-carbohydrate liquid diet (HCD)), high fat (5% EtOH + high-fat liquid diet (HFD)), and high protein (5% EtOH + high-protein liquid diet (HPD))) and subject to the indicated conditions for 6 weeks. All animal experiments were performed according to National Institutes of Health Guidelines for the Care and Use of Laboratory Animals and approved by the Institutional Animal Care and Use Committee of the College of Medicine, Hallym University (Hallym2016-4) (Figure 1A and Table 1).

For cognitive function, we performed T-maze studies 3 times altogether before and after alcohol binge and following the next day. In addition, alcohol binge was provided at 5 g/kg/per 15 days, 40% ethanol. We used a T-shape maze constructed of acrylic with opaque sides, and the maximum time was set at 60 s. Before alcohol binging, all animals were pretrained to searching the normal chow diet toward the left side blocking one direction. After alcohol binging, we measured the time that mice search for the normal chow diet (Figure 1B). To assess working memory, we conducted pretraining and poststudy assessments. At the pretrial stage, the mice were trained to run from the start to receive a food reward. We trained mouse until the mice found food before starting the test. For the evaluation of the effect of alcohol on cognition, we performed the test before and after alcohol binging at 11, 25, and 44 days during study. In addition, we measured the mice performance time in each diet group (HCD, HPD, HFD).

The mice were euthanized with ether anesthesia. A midline abdominal incision was created, and blood was collected at the heart. Livers were rapidly resected for pathological assessment and stored at −80 °C for the evaluation of liver enzyme profiles.

### 2.4. Pathology 

Specimens were fixed with 10% formalin and routinely embedded in paraffin; the tissue sections were processed with hematoxylin and eosin, Masson’s trichrome, and reticulin fiber staining. Fatty liver was classified by the clinical research network scoring system [16]. Fatty liver was classified from grade 0 to grade 3 (0: <5%, 1: 5–33%, 2: 34–66%, and 3: >66% of steatosis). Inflammation was classified from grades 0 to 3 (0: none, 1: 1–2 foci per ×20 field, 2: 2–4 foci per ×20 field, and 3: >4 foci per ×20 field). We commissioned a pathological report to the pathology department of a Chuncheon Sacred Heart Hospital. All specimens were blindly analyzed by 1 hepato-pathologist.

### 2.5. Western Blot Analysis of the Brain

The whole brain was washed with ice-cold PBS and lysed with a modified RIPA buffer containing 50 mM Tris-HCl pH 7.4, 150 mM NaCl, 1% Triton X-100, 0.1% sodium dodecyl sulfate (SDS), 0.5% sodium deoxycholate, 1 mM ethylenediaminetetraacetic acid (EDTA), protease inhibitors (Pierce Biotechnology, Rockford, IL, USA), 1 mM Na_3 VO4_, and 1 mM NaF. Brain homogenates were centrifuged at 15,000× *g* for 15 min at 4 °C, and the protein concentrations in the supernatants were analyzed using a BCA Protein Assay Kit (Thermo Fisher Scientific, Waltham, MA, USA). Equal amounts of proteins (40 μg/lane) were separated using sodium dodecyl sulfate-polyacrylamide electrophoresis, transferred onto 0.45-μm pore polyvinylidene fluoride (PVDF) membranes (Merck Millipore, Lake Placid, NY, USA), and blocked with 5% skim milk in 1 × PBS containing 0.1% Tween 20 (PBST) for 1 h at room temperature. The following primary antibodies were added to the membranes, which were incubated overnight at 4 °C: Anti-Synapsin, anti-Synaptophysin, anti-SNAP25, anti-PSD95, anti-Synaptotagmin (Abcam, Cambridge, MA, USA), anti-Vamp2 (Cell Signaling Technology, Beverly, MA, USA), anti-β-actin (Sigma-Aldrich, St. Louis, MO, USA). The membranes were washed with PBST thrice for 10 min each and then incubated with the following secondary antibodies for 1 h: Goat anti-mouse IgG or goat anti-rabbit IgG (Thermo Fisher Scientific) conjugated with horseradish peroxidase (HRP). The immunoreactive bands were visualized on digital images captured with an ImageQuantTM LAS4000 imager (GE Healthcare Life Sciences, Piscataway, NJ, USA) using the EzwestLumi plus Western blot detection reagent (ATTO Corporation, Tokyo, Japan). The band intensities were quantified using the ImageJ (NIH) program. Statistical analyses were performed using GraphPad Prism 4 (San Diego, CA, USA).

## 3. Results

### 3.1. Patient Characteristics

The proportion of male was 82% (14/17) and 77% (20/26) in the BMI < 22 group and BMI ≥ 22 group, respectively. The mean age of patients was 53.2 ± 10.2 and 50.0 ± 8.0 years. The mean BMIs of patients were 21.9 ± 4.6 and 24.9 ± 3.0 in two groups (BMI < 22 and BMI ≥ 22) (*p* < 0.01). The period of education and blood level of hemoglobin, total bilirubin, AST, AlT, ALP, and γ-GT did not exhibit significant differences (Table 2).

### 3.2. Body Mass Index and Cognitive Function

In the comparison of cognitive function between female and male, repetition (13.5 ± 2.4 and 14.9 ± 0.4) and naming K-BNT (44.7 ± 7.1 and 51.9 ± 6.5) tests showed significant different. The neuropsychological tests such as digit span forward score (6.7 ± 1.8 and 7.5 ± 1.6), stroop tests (color reading time per items (24.2 ± 26.5 and 43.6 ± 32.4) and interference score (33.9 ± 31.9 and 52.3 ± 33.9)), K-MMSE (language, 7.4 ± 1.4 and 7.9 ± 0.4), and K-BNT language (11.8 ± 2.7 and 13.0 ± 1.8) revealed high scores in patients with BMI ≥ 22 (*p* < 0.05). Other neuropsychological tests did not exhibit significant differences between two groups (*p* > 0.05) (Table 3 and Appendix A).

### 3.3. Changes in the Body Weight and Pathology

In the comparison of body weight, HPD and HCD groups (24.5 ± 2.0 and 24.2 ± 1.3) weighed more than the alcohol group (22.7 ± 1.3). About 8% and 7% increase in weight was shown in HPD and HCD groups. The diet of both groups recovered weight loss caused by alcohol. In the analysis of the liver/body weight ratio (%), the HFD group (5.60 ± 0.40) exhibited significant changes compared with the alcohol group (4.52 ± 0.31) (*p* < 0.01). The HPD (5.10 ± 1.04) and HCD (4.85 ± 0.44) groups did not exhibit differences in the liver/body weight ratio (Figure 2A).

In the pathological results, each group did not exhibit differences in the inflammation grade. The HPD (2.3 ± 0.5) and HFD (2.0 ± 0.5) groups exhibit significant increases in steatosis compared with the alcohol group (1.5 ± 0.7) (Figure 2B). Diets in both groups did not produce hepatitis, but caused an increase in liver fat.

### 3.4. Cognitive Function Based on Diet Groups

In this study, alcoholic liver disease occurred after 25 days’ alcohol diet (Figure 3A). The reduction in cognitive function was proportional to the period of alcohol administration. As measured on the 11th, 25th, and 44th day in the alcohol group, the T-maze time was instantly reduced after binge drinking alcohol. Interestingly, temporary recovery of cognitive function occurred after binging, but temporary improvement in cognitive function declined again the day after binge 3B (Figure 3B). In the posthoc analysis of alcohol group (11 days, 25 days, and 44 days), all comparisons showed significant difference [mean times of escape latency (standard deviation) are 34 (8), 48 (5), and 58 (6) seconds in before alcohol binge, 21 (6), 26 (5), and 29 (4) in after alcohol binge, and 36 (11), 42 (7), and 52 (5) in 1 day after binge, respectively). Comparing groups according to diet, cognitive function improved in the HCD and HFD groups compared with the alcohol group. In particular, the HCD group significantly reduced the time proportional to the duration of administration (Figure 3C,D).

### 3.5. Brain Markers in the High-Protein Diet Group

The levels of the synapsin-1, synaptophysin, SNAP25, PSD95, synaptotagmin, and Vamp2 were measured by band quantification and normalized with the levels of B-actin. Synapsin-1, synaptophysin, and SNAP25 represent presynaptic markers. PSD95 is postsynaptic marker. Synaptotagmin and Vamp2 represent vesicle markers. Vesicles function as pouches that store and transport substances within cells and digest cellular products.

Synapses exhibit mechanical and functional properties, transmit signals, and process information in the brain. Synapses are responsible for overall brain function and is also related to cognitive function [17,18]. To evaluate synaptic function, we conducted Western blotting using samples from the HPD group and alcohol-induced mice. The result shows that synaptic function did not change in the alcohol and HPD groups (Figure 4). Summarizing the results of brain tissue, changes in brain protein and function do not occur due to alcohol diet.

## 4. Discussion

BMI has been used as an index of an individual’s fat levels and nutritional state in various disease. Accordingly, a high BMI is related with brain atrophy and cognitive dysfunction in some brain diseases [19,20,21,22]. Regarding ALD, a negative relationship is noted between alcohol consumption and BMI [23]. Most patients have a low BMI index due to insufficient calorie intake. Few studies are available on the association between cognitive dysfunction and BMI in alcoholic liver disease. In this study, patients (BMI ≥ 22) with cirrhosis exhibited improved cognitive functions, such as visuospatial function, Stroop, attention, MMSE, and language tests, compared with patients with low BMI (<22). Therefore, we suggest that BMI is closely related with cognitive function in ALD patients. 

Chronic alcohol consumption results in brain damage, especially in the frontal lobe [24]. Alcohol use disorder is related to chronic impairment in emotion recognition and reasoning [25]. Problematic alcohol drinking activity is correlated with an increased risk of cognitive dysfunction [26]. Other report suggested that consumption of a small amount of alcohol may reduce the risk of cognitive dysfunction in woman [27]. ALD severity is dependent on the amount and duration of the alcohol intake [28]. In our T-maze test results utilizing male mice, chronic alcohol intake decreased cognitive function. Taken together, chronic alcohol intake affects cognitive function, but gender differences exist.

Patients with compensated liver cirrhosis does not show symptom about abnormal cognitive function. It can be diagnosed with a cognitive function test [29]. Patients with compensated liver cirrhosis often do not receive medical treatment due to no symptoms, but they show abnormalities such as traffic accidents or memory impairment in real life. Therefore, as shown in the results of this study, it is important to maintain nutritional status to restore cognitive function and supplement diet.

T-maze test has been used for the function with memory and spatial learning in animal model [30,31]. Different sizes and shapes of T-maze test might be used according to the types of animal model. In our animal model of alcohol-induced liver disease, one practice of reaching food was performed prior to the T maze test.

An interesting fact is that mice exhibited a temporarily reduced T-maze time after alcohol binging in all groups. In our results, the reduced time in T-maze tests after binging worsened after one day. These result are consistent with a recent clinical trial demonstrating that alcohol improved creative problem solving but not divergent thinking [32]. Another study demonstrated that moderate alcohol drinking reduced working memory capacity but increased performance [33]. Improving of performance and abnormal pleasure is a symptom related to addiction, and it is probably one of the reasons why patients continue drinking alcohol. This result indicates that chronic alcohol intake may lead to temporary improvements in cognitive function but ultimately leads to fatal damage. As a result, abstinence is the most important treatment for patients with alcoholic liver disease.

In our results, a high-fat diet yielded the shortest time to find food in the alcohol animal model. HPD increased the time required to find food in the T-maze test. In a previous report, a high-fat diet improved neuro-inflammation and neurogenesis [34]. In addition, a high-fat diet increased lipid accumulation in the cortex and brain sensitivity [35]. Regarding obesity, high-fat diet-induced obesity caused cognitive dysfunction. In this study, the results were different because the effects of diet were observed in alcohol models and not the obesity model. Dietary studies on alcoholic liver disease are limited, suggesting that the HFD in this study serves as a dietary regimen for improving cognitive function in alcoholic liver disease. In this study, dextrin was added instead of fat to standardize caloric intake. 

Regarding HCD-related improvement in cognition, we hypothesize that maltose dextrin might increase memory. Another clinical report demonstrated that memory is facilitated by offering lecithin, carnitine, and glucose supplements [36]. A low-carbohydrate diet aggravated attention and induced confusion in a previous report [37]. Taken together, HFD or HCD can be used as nutritional therapy in patients with ALD. 

Alcohol intoxication reduces brain activity in the cortical and subcortical regions, including the temporal lobe, which contains the hippocampus [38]. In cirrhosis and advanced ALD, atrophic changes in the brain are dominant [9], but functional changes take precedence over structural changes in alcohol use disorder. In our data, pathological examination revealed hepatitis not cirrhosis and did not reveal differences in synapsin1, synaptophysin, SNAP25, PSD95, synaptotagmin, and Vamp2 protein levels in brain. These results suggest that alcohol intake does not result in structural issues in the brain but does reduce cognitive function. In addition, nutritional therapy for ALD should be started at an early stage before brain changes occur.

Our results from an alcohol-induced model demonstrated that a high-fat diet improves working memory. However, the limitation of this study is that HFD is not related to steatosis levels. In addition, in Western blot analyses, synaptic functions that have a direct or indirect effect on all processes in the overall brain exhibited no significant differences in the protein diet and alcohol group.

In summary, neuropsychological tests of ALD patients revealed that high BMI patients exhibited relatively increased cognitive function compared with low BMI patients. Similarly, in an alcohol-induced animal model, the high-fat diet group, which consumed enough calories, exhibited improved working memory performance. Further studies are need to explore whether alcohol consumption enhances cognitive function.

## Figures and Tables

**Figure 1 nutrients-13-00185-f001:**
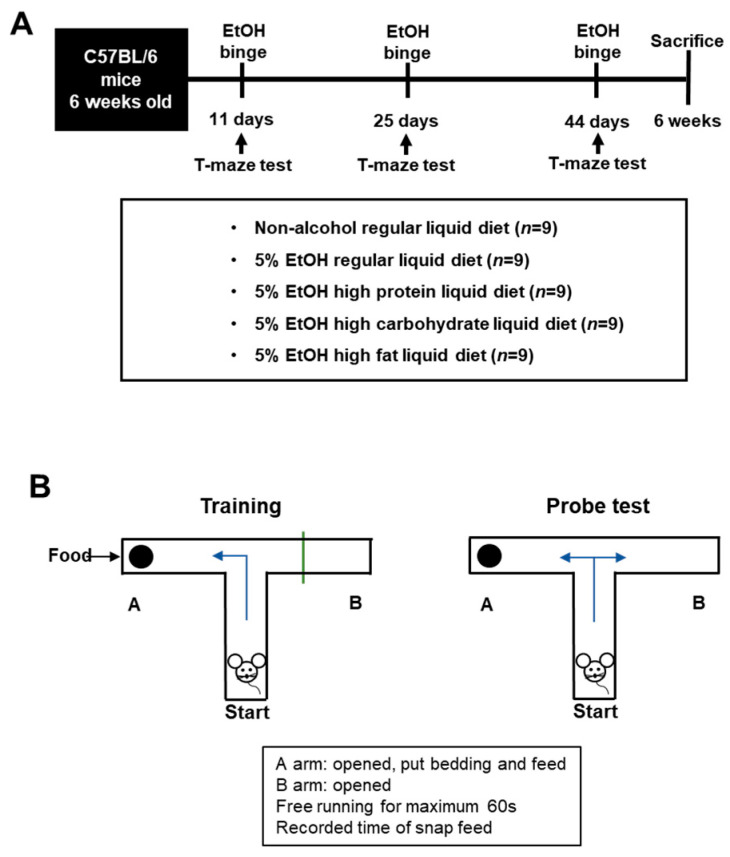
Animal experiment. (**A**). Mice were sub-divided into five groups (*n* = 9/group; normal regular liquid diet, control (5% EtOH liquid diet + regular liquid diet), high carbohydrate (5% EtOH + HCD), high fat (5% EtOH + HFD), and high protein (5% EtOH + HPD)) and subject to the indicated conditions for six weeks. (**B**). Schematic of T-maze appliance for working memory study.

**Figure 2 nutrients-13-00185-f002:**
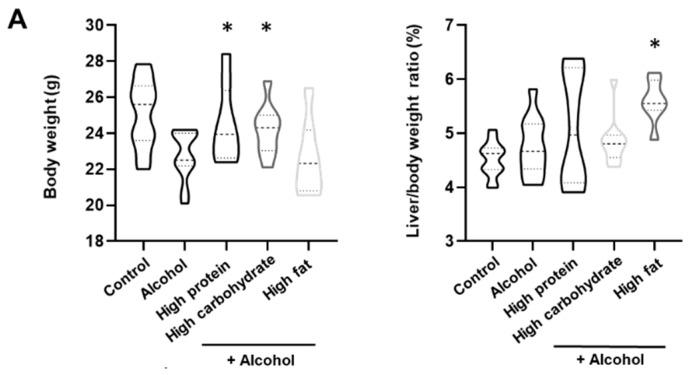
Pathological result according to diet group. (**A**). Liver/Body weight ratio (%), violin plot with min and max score (thick dotted line (median) and thin dot line (interquartile range)), (**B**). Pathological result of diet group. * *p* < 0.05 difference compared with alcohol group. The data are presented as the mean ± standard error of the mean (SEM) and statistically analyzed using a Mann–Whitney test.

**Figure 3 nutrients-13-00185-f003:**
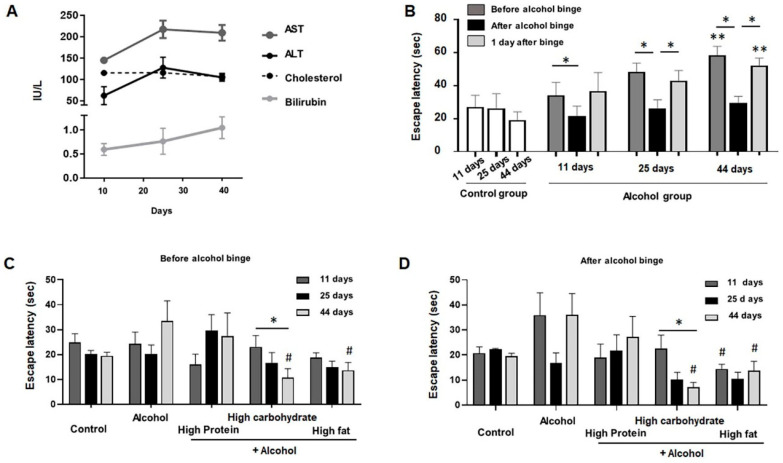
Liver enzyme and T-maze test. (**A**). The serum level of liver enzyme in alcohol group mice (*n* = 20). (**B**). Change of T-maze test in alcohol mice (*n* = 20). (**C**). T-MAZE before alcohol binge. (**D**). T-MAZE after alcohol binge. The data are presented as the mean ± standard error of the mean (SEM) and were statistically analyzed using a one-way ANOVA (LSD) and Mann–Whitney test. * *p* < 0.05 difference at the posthoc test (LSD); ** *p* < 0.05 difference between the three groups (11 days, 25 days, and 44 days); # *p* < 0.05 difference compared with alcohol group. AST, (Aspartate aminotransferase); ALT, (Alanine aminotransferase)

**Figure 4 nutrients-13-00185-f004:**
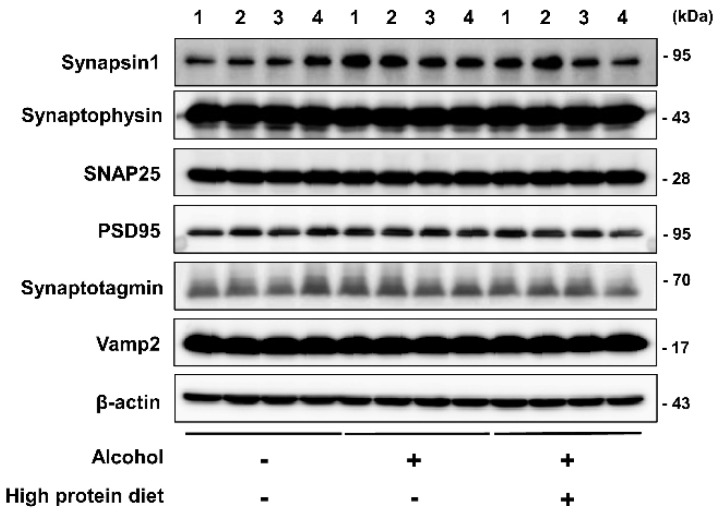
Expression of synaptic neuron in alcohol induced high protein diet group. Representative Western blot of mouse hippocampus homogenate.

**Table 1 nutrients-13-00185-t001:** Nutrient composition of high protein, high carbohydrate, and high fat liquid diet.

Diet	Regular	High Protein	High Carbohydrate	High Fat
Ingredient	g/L	g/L	g/L	g/L
Casein	41.4	57.6	41.4	41.4
DL-methionine	0.3	0.4	0.3	0.3
L-cystine	0.5	0.65	0.5	0.5
Cellulose	10	10	10	10
Maltose dextrin	25.6	66	83.6	24.05
Corn oil	8.5	2.5	2.5	31.1
Olive oil	28.4	8.4	8.4	28.4
Safflower oil	2.7	2.7	2.7	2.7
Mineral mix	8.75	8.75	8.75	8.75
Vitamin mix	2.5	2.5	2.5	2.5
Choline bitartrate	0.53	0.53	0.53	0.53
Xanthan gum	3	3	3	3

**Table 2 nutrients-13-00185-t002:** Baseline characteristics of patient with compensated alcoholic cirrhosis.

Variables (Mean)	BMI < 22 (*n* = 17)	BMI ≥ 22 (*n* = 26)	*p* Value
Male *n* (%)	14 (82)	20 (77)	
Age	53.2 (10.2)	50.0 (8.0)	0.659
BMI	21.9 (4.6)	24.9 (3.0)	0.001
Education period (years)	9.3 (4.0)	10.8 (4.0)	0.302
Hemoglobin (g/dL)	11.7 (2.0)	12.4 (2.5)	0.368
Albumin (g/dL)	3.8 (0.6)	3.6 (0.8)	0.298
Total bilirubin (mg/dL)	0.9 (0.5)	1.7 (1.3)	0.023
AST (IU/L)	54.6 (38.7)	69.7 (50.8)	0.302
ALT (IU/L)	25.0 (17.8)	36.8 (30.0)	0.151
ALP (IU/L)	129.5 (47.7)	135.8 (62.5)	0.726
γ-GT (IU/L)	434.7 (455.2)	333.0 (382.1)	0.434

AST, Aspartate aminotransferase; ALT, Alanine aminotransferase; ALP, alkaline phosphatase; γ-GT, gamma-glutamyl transpeptidase.

**Table 3 nutrients-13-00185-t003:** Neuropsychological tests related cognitive function.

Variable	Normal Control (*n* = 1000)	BMI < 22 (*n* = 17)	BMI ≥ 22 (*n* = 26)	*p*-Value
Visuospatial function				
Digit span forward	7.7 (1.4)	6.7 (1.8)	7.5 (1.6)	0.039
Digit span backward	4.3 (1.0)	3.8 (1.3)	4.3 (1.5)	0.138
Stroop test				
Color reading correct	85.0 (10)	72.4 (27.1)	83.8 (26.0)	0.079
Color reading time	120.0 (5.0)	119.3 (3.4)	116.4 (9.4)	0.076
Color reading correct response rate	1.0 (0.1)	1.0 (0.1)	1.0 (0.1)	0.432
Color reading time per items	45.1 (20.2)	24.2 (26.5)	43.6 (32.4)	0.006
Interference scores	55.5 (20.0)	33.9 (31.9)	52.3 (33.9)	0.017
K-MMSE				
K-MMSE, language	7.8 (0.5)	7.4 (1.4)	7.9 (0.4)	0.037
Language				
K-BNT	13.1 (1.3)	11.8 (2.7)	13.0 (1.8)	0.017

RCFT, Ray-complex figure test; K-MMSE, Korea-mini mental status examination; K-BNT, Korean version-Boston naming test.

## Data Availability

The data presented in this study are available on request from the corresponding author.

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
