# Peer review of "Nutritional Status and Diet Style Affect Cognitive Function in Alcoholic Liver Disease"

_nutrients, 2021, doi:10.3390/nu13010185_

Round 1
Reviewer 1 Report
In this manuscript the authors explored the effect of nutritional state and diet on cognitive function in alcoholic liver disease. The topic is of great interest but the writing needs to be improved. In my opinion, the results must be written in a clearer and more detailed way, especially those concerning the animal model.
The composition of the different diets administered to animal model should be reported. The human study missing a healthy control. Did the authors find gender differences in the various parameters examined? Please report a table showing the results by gender. The statistical analyses performed must be described in the text of the manuscript.
Author Response
nutrients-1043290
“Nutritional status and diet style affect cognitive function in alcoholic liver disease”
Point-to-point responses to comments by the Reviewer 1
First of all, we would like to thank the Reviewer1 for his/her comments, which helped us to improve this manuscript.
- Comment 1: In this manuscript the authors explored the effect of nutritional state and diet on cognitive function in alcoholic liver disease. The topic is of great interest but the writing needs to be improved. In my opinion, the results must be written in a clearer and more detailed way, especially those concerning the animal model.
- Response 1: We appreciate the Reviewer’s thoughtful comment. We revised results section of manuscript. And, we carefully checked the results of the two groups.
“The proportion of male was 82% (14/17) and 77% (20/26) in the BMI < 22 group and BMI ≥ 22 group, respectively. The mean age of patients was 53.2 ± 10.2 and 50.0 ± 8.0 years. The mean BMIs of patients were 21.9 ± 4.6 and 24.9 ± 3.0 in two groups (BMI < 22 and BMI ≥ 22) (p < 0.01). The period of education and blood level of hemoglobin, total bilirubin, AST, AlT, ALP, and γ-GT did not exhibit significant differences (Table 2).”
“The neuropsychological tests such as digit span forward score (6.7 ± 1.8 and 7.5 ± 1.6), stroop tests (color reading time per items [24.2 ± 26.5 and 43.6 ± 32.4] and interference score [33.9 ± 31.9 and 52.3 ± 33.9]), K-MMSE (language, 7.4 ± 1.4 and 7.9 ± 0.4), and K-BNT language (11.8 ± 2.7 and 13.0 ± 1.8) revealed high scores in patients with BMI ≥ 22 (p < 0.05). Other neuropsychological tests did not exhibit significant differences between two groups (p > 0.05) (Table 3 and supplementary Table 1).”
“In the comparison of body weight, HPD and HCD groups weighed more than the alcohol group. The diet of both groups recovered weight loss caused by alcohol. In the analysis of the liver/body weight ratio (%), the HFD group (5.60 ± 0.40) exhibited significant changes compared with the alcohol group (4.52 ± 0.31) (p < 0.01). The HPD (5.10 ± 1.04) and HCD (4.85 ± 0.44) groups did not exhibit differences in the liver/body weight ratio (Fig. 2A).”
“In this study, alcoholic liver disease occurred after 25 days’ alcohol diet (Fig. 3A). The reduction in cognitive function was proportional to the period of alcohol administration. As measured on the 11th, 25th, and 44th day in the alcohol group, the T-maze time was instantly reduced after binge drinking alcohol. Interestingly, temporary recovery of cognitive function occurred during binging, but cognitive function resumed one day after binge (Fig. 3B). Comparing groups according to diet, cognitive function improved in the HCD and HFD groups compared with the alcohol group. In particular, the HCD group significantly reduced the time proportional to the duration of administration (Fig. 3C and D).”
“Summarizing the results of brain tissue, changes in brain protein and function do not occur due to alcohol diet.”
- Comment 2: The composition of the different diets administered to animal model should be reported. The human study missing a healthy control. Did the authors find gender differences in the various parameters examined? Please report a table showing the results by gender. The statistical analyses performed must be described in the text of the manuscript.
- Response 2: We agree with the reviewer’s comment and apologize for causing confusion. We appreciate for the further evaluation. We added Table 1 which includes all diet information. Regarding healthy control in human data, we included control group’s data in the Table 3.
In the analysis of cognitive function between female and male, repetition (13.5 ± 2.4 and 14.9 ± 0.4) and naming K-BNT (44.7 ± 7.1 and 51.9 ± 6.5) tests showed significant different. We mentioned that in the result section. Because of lack of female patients, we did not do the subgroup analysis
“In the comparison of cognitive function between female and male, repetition (13.5 ± 2.4 and 14.9 ± 0.4) and naming K-BNT (44.7 ± 7.1 and 51.9 ± 6.5) tests showed significant different.”
|
Diet |
Regular |
High protein |
High carbohydrate |
High fat |
|
|
Ingredient |
g/Liter |
g/Liter |
g/Liter |
g/Liter |
|
|
Casein |
41.4 |
57.6 |
41.4 |
41.4 |
|
|
DL-methionine |
0.3 |
0.4 |
0.3 |
0.3 |
|
|
L-cystine |
0.5 |
0.65 |
0.5 |
0.5 |
|
|
Cellulose |
10 |
10 |
10 |
10 |
|
|
Maltose dextrin |
25.6 |
66 |
83.6 |
24.05 |
|
|
Corn oil |
8.5 |
2.5 |
2.5 |
31.1 |
|
|
Olive oil |
28.4 |
8.4 |
8.4 |
28.4 |
|
|
Safflower oil |
2.7 |
2.7 |
2.7 |
2.7 |
|
|
Mineral mix |
8.75 |
8.75 |
8.75 |
8.75 |
|
|
Vitamin mix |
2.5 |
2.5 |
2.5 |
2.5 |
|
|
Choline bitartrate |
0.53 |
0.53 |
0.53 |
0.53 |
|
|
Xanthan gum |
3 |
3 |
3 |
3 |
|
Table 1. Nutrient composition of high protein, high carbohydrate, and high fat liquid diet
|
Variable |
Normal control (n=1,000) |
BMI < 22 (n=17) |
BMI ≥ 22 (n=26) |
P- value* |
|
|
|
Visuospatial function |
|
|
|
|
|
|
|
Digit span forward |
7.7 (1.4) |
6.7 (1.8) |
7.5 (1.6) |
0.039 |
|
|
|
Digit span backward |
4.3 (1.0) |
3.8 (1.3) |
4.3 (1.5) |
0.138 |
|
|
|
Stroop test |
|
|
|
|
|
|
|
Color reading correct |
85.0 (10) |
72.4 (27.1) |
83.8 (26.0) |
0.079 |
|
|
|
Color reading time |
120.0 (5.0) |
119.3 (3.4) |
116.4 (9.4) |
0.076 |
|
|
|
Color reading correct response rate |
1.0 (0.1) |
1.0 (0.1) |
1.0 (0.1) |
0.432 |
|
|
|
Color reading time per items |
45.1 (20.2) |
24.2 (26.5) |
43.6 (32.4) |
0.006 |
|
|
|
Interference scores |
55.5 (20.0) |
33.9 (31.9) |
52.3 (33.9) |
0.017 |
|
|
|
K-MMSE |
|
|
|
|
|
|
|
K-MMSE, language |
7.8 (0.5) |
7.4 (1.4) |
7.9 (0.4) |
0.037 |
|
|
|
Language |
|
|
|
|
|
|
|
K-BNT |
13.1 (1.3) |
11.8 (2.7) |
13.0 (1.8) |
0.017 |
|
|

Reviewer 2 Report
In the manuscript entitled “Nutritional Status and Diet Style Affect Cognitive Function in Alcoholic Liver Disease”, the authors assessed the effect of BMI and diet on cognitive function in patients with alcoholic liver disease. Dietary effect om cognitive function was also investigated in C57Bl/6J mice models of alcoholic liver disease (fed 5% ethanol+different liquid diets for 8 weeks) with effects on binge drinking being also assessed. The work is interesting; however, methodology details are missing and the results need to be clearer with a more in-depth presentation of the article overall. My comments are listed below.
-Line 101, please specify the neurology specialist and hepato-pathologist ………it is not specified under “Author Contributions”
- figure 1A, shows that animals were sacrificed 6 weeks after the study was initiated, is it 6 or 8 weeks? Also, was the max time in the maze 60 (figure 1B) or 90 sec (line 118)
- please specify diet sources and composition. Also, how alcohol binging was performed.
- line 116 “For cognitive function, we performed T-maze studies weekly before and after alcohol binging” there are 2 measurements after binging (figure line 3B), one of them 1 day after binging, when was the other measurement taken?
- line 118, “. Before alcohol binging, all animals were pretrained to searching the normal chow diet toward the left side blocking one direction” please describe how long were the mice trained for and the frequency of training
-line 123, “For the evaluation of the effect of alcohol on cognition, we performed the test before and after alcohol binging at 11, 25, and 44 days during study”, when was the before measurement taken? Are these time points correct? Some are not in line with figure 1A….please clarify/correct
- line 123, “In addition, we measured the mice performance time in each diet group (HCD, HPD, HFD)” these groups had chronic alcohol (5%) in their diet a well, right?
- lines 167-171, “The neuropsychological tests of 43 patients, including visuospatial functions by digit span forward score (6.7 ± 1.8 and 7.5 ± 1.6), K-MMSE (language, 7.4 ± 1.4 and 7.9 ± 0.4), language score K BNT (11.8 ± 2.7 and 13.0 ± 1.8), attention by RCFT copy score (72.1 ± 25.9 and 58.4 ± 33.6) and Stroop test (color reading time per items [24.2 ± 26.5 and 43.6 ± 32.4] and interference score [33.9 ± 31.9 and 171 52.3 ± 33.9]), revealed high scores in patients with BMI ≥ 22 (p < 0.05).” actually RCFT copy score was reduced, please comment.
- Figure 2A, what do these shapes and their internal lines represent? Mean, quartiles?
- Figure 2, please state the statistical test used for assessing the differences in the legend. Also state how was the inflammation grade assessed (add this to the methods section)
- Figure 2b, shouldn’t the +alcohol line be extended to the high protein diet as well??
- please refer to fig 2 & 3 panels in the text under “Changes in the Body Weight and Pathology” & “Cognitive Function Based on Diet Groups”, respectively. Also, state % improvement in body weight in the HPD and HCD
- Figure 3A isn’t discussed at all, you are showing only levels in the alcohol group. Did you measure liver enzymes in the rest of the groups?
- figure 3b, please add the y-axis label
- section 3.4 under results is confusing and needs to be rewritten clearly
- figure 3, control and alcohol readings are different among different panels although they should be the same. For example, panel b the 10 and 25 days control bars are close and higher than the 40 days bar but this isn’t the case in panels c and d. the same for alcohol measurements before and after binge. Please clarify. Also state the statistical test used for comparison in the legend
Author Response
nutrients-1043290
“Nutritional status and diet style affect cognitive function in alcoholic liver disease”
Point-to-point responses to comments by the Reviewer 2
First of all, we would like to thank the Reviewer2 for his/her comments, which helped us to improve this manuscript.
Specific Comments:
- Comment 1: Line 101, please specify the neurology specialist and hepato-pathologist ………it is not specified under “Author Contributions”
- Response 1: We appreciate the Reviewer’s thoughtful comment. When we designed our study, the pathology was paid to the company and commissioned. Regarding about neurology specialist, neurological examination was also performed with payment of research fee. All reports and results So, we did not mention about pathological specialist.
- Comment 2: figure 1A, shows that animals were sacrificed 6 weeks after the study was initiated, is it 6 or 8 weeks? Also, was the max time in the maze 60 (figure 1B) or 90 sec (line 118)
- Response 2: We apologize for causing confusions. We eliminated “maximum time 90s” and “8 weeks” replaced it with “maximum time 60s” and “6 weeks”
- Comment 3 and 4: please specify diet sources and composition. Also, how alcohol binging was performed. “For cognitive function, we performed T-maze studies weekly before and after alcohol binging” there are 2 measurements after binging (figure line 3B), one of them 1 day after binging, when was the other measurement taken?
- Response 3 and 4: We appreciate the Reviewer’s thoughtful comment. We thought that alcohol consumption had an effect instantly on cognitive function. Therefore, 1 day after, we measured T-maze test one more for normal state. As the reviewer mentioned that the sentence "For cognitive function, we performed T-maze studies weekly before and after alcohol binging" can cause confusion by measuring 2 times compared to fig 3B. So, we revised and added the following clarification in the "Experimental Animals" section: "For cognitive function, we performed T-maze studies 3 times before alcohol binge, after alcohol binge and 1 day after. Also, alcohol binge was provided at 5 g/kg/per 15 days, 40% ethanol." Additionally, we included nutrient sources and composition with table1.
Table 1. Nutrient composition of high protein, high carbohydrate, and high fat liquid diet
|
Diet |
Regular |
High protein |
High carbohydrate |
High fat |
|
|
Ingredient |
g/Liter |
g/Liter |
g/Liter |
g/Liter |
|
|
Casein |
41.4 |
57.6 |
41.4 |
41.4 |
|
|
DL-methionine |
0.3 |
0.4 |
0.3 |
0.3 |
|
|
L-cystine |
0.5 |
0.65 |
0.5 |
0.5 |
|
|
Cellulose |
10 |
10 |
10 |
10 |
|
|
Maltose dextrin |
25.6 |
66 |
83.6 |
24.05 |
|
|
Corn oil |
8.5 |
2.5 |
2.5 |
31.1 |
|
|
Olive oil |
28.4 |
8.4 |
8.4 |
28.4 |
|
|
Safflower oil |
2.7 |
2.7 |
2.7 |
2.7 |
|
|
Mineral mix |
8.75 |
8.75 |
8.75 |
8.75 |
|
|
Vitamin mix |
2.5 |
2.5 |
2.5 |
2.5 |
|
|
Choline bitartrate |
0.53 |
0.53 |
0.53 |
0.53 |
|
|
Xanthan gum |
3 |
3 |
3 |
3 |
|
- Comment 5: Before alcohol binging, all animals were pretrained to searching the normal chow diet toward the left side blocking one direction” please describe how long were the mice trained for and the frequency of training
- Response 5: Thanks for raising this point. We trained only 1 time. We mentioned this at 2.3. Experimental Animals section.
“We trained mice once until the mouse found food.”
- Comment 6, 7 and 14: For the evaluation of the effect of alcohol on cognition, we performed the test before and after alcohol binging at 11, 25, and 44 days during study”, when was the before measurement taken? Are these time points correct? Some are not in line with figure 1A….please clarify/correct. “In addition, we measured the mice performance time in each diet group (HCD, HPD, HFD)” these groups had chronic alcohol (5%) in their diet a well, right?” “figure 3b, please add the y-axis label.”
- Response 6, 7 and 14: In figure 1A and figure 3B, we clarified the T-maze performed date.
We gave chronic alcohol in diet group (HCD, HPD, HFD). Authors can see this fact in the Fig. 1A. We changed “10 days, 40 days” → “11 days, 44 days” And in figure 3B, we added the y-axis label.
Original figure 1A
Revised figure 1A
Original figure 3B
Revised figure 3B
- Comment 8: “The neuropsychological tests of 43 patients, including visuospatial functions by digit span forward score (6.7 ± 1.8 and 7.5 ± 1.6), K-MMSE (language, 7.4 ± 1.4 and 7.9 ± 0.4), language score K BNT (11.8 ± 2.7 and 13.0 ± 1.8), attention by RCFT copy score (72.1 ± 25.9 and 58.4 ± 33.6) and Stroop test (color reading time per items [24.2 ± 26.5 and 43.6 ± 32.4] and interference score [33.9 ± 31.9 and 171 52.3 ± 33.9]), revealed high scores in patients with BMI ≥ 22 (p < 0.05).” actually RCFT copy score was reduced, please comment.
- Response 8: We agree with the reviewer’s comment and apologize for mistakes. We removed RCFT copy score from the table and results.
- Comment 9: Figure 2A, what do these shapes and their internal lines represent? Mean, quartiles?
- Response 9: This graph is violin plot and we mentioned this plot on the Figure 2A.
“violin plot with min and max score (thick dotted line [median] and thin dot line [interqartile range])”
- Comment 10: Figure 2, please state the statistical test used for assessing the differences in the legend. Also state how was the inflammation grade assessed (add this to the methods section)
- Response 10: Thanks for these suggestions. We included the following sentence in the “Pathology” section: “Inflammation was classified from grades 0 to 3 (0: none, 1: 1–2 foci per ×20 field, 2: 2–4 foci per ×20 field, and 3: >4 foci per ×20 field).”
- Comment 11: shouldn’t the +alcohol line be extended to the high protein diet as well??
- Response 11: As reviewer suggested, we extended to the high protein diet on Fig. 2.
- Comment 12: please refer to fig 2 & 3 panels in the text under “Changes in the Body Weight and Pathology” & “Cognitive Function Based on Diet Groups”, respectively. Also, state % improvement in body weight in the HPD and HCD
- Response 12: Thank you for raising this point. In “Changes in the Body Weight and and Pathology” we referred to fig A and B per sentence. Also, in “Cognitive Function Based on Diet Groups”, we referred figure 3 panels according to each sentence appropriately.
“In the comparison of body weight, HPD and HCD groups (24.5 ± 2.0 and 24.2 ± 1.3)weighed more than the alcohol group (22.7 ± 1.3). About 8% and 7% increase in weight was shown in HPD and HCD groups.”
- Comment 13: Figure 3A isn’t discussed at all, you are showing only levels in the alcohol group. Did you measure liver enzymes in the rest of the groups?
- Response 13: We agree with the reviewer’s comment. We mentioned about Fig 3A. “In this study, alcoholic liver disease occurred after 25 days’ alcohol diet (Fig. 3A).”
- Comment 15: section 3.4 under results is confusing and needs to be rewritten clearly
- Response 15: We apologize for confusion about the lack of explanation.
“In this study, alcoholic liver disease occurred after 25 days’ alcohol diet (Fig. 3A). The reduction in cognitive function was proportional to the period of alcohol administration. As measured on the 11th, 25th, and 44th day in the alcohol group, the T-maze time was instantly reduced after binge drinking alcohol. Interestingly, temporary recovery of cognitive function occurred during binging, but cognitive function resumed one day after binge (Fig. 3B). Comparing groups according to diet, cognitive function improved in the HCD and HFD groups compared with the alcohol group. In particular, the HCD group significantly reduced the time proportional to the duration of administration (Fig. 3C and D).”
- Comment 16: figure 3, control and alcohol readings are different among different panels although they should be the same. For example, panel b the 10 and 25 days control bars are close and higher than the 40 days bar but this isn’t the case in panels c and d. the same for alcohol measurements before and after binge. Please clarify. Also state the statistical test used for comparison in the legend
- Response 16: We agree with the reviewer’s comment and apologize for causing confusion regarding the among different panels of control and alcohol groups. In Figure 3B, we showed the average values for each day before binge drinking, after binge drinking and 1 day after binge drinking. Likewise, Alcohol group ~
As per the reviewer’s request, we included this sentence in the fig3 legend section: “The data are presented as the mean ± standard error of the mean (SEM) and were statistically analyzed using a one-way ANOVA(?).”
Figure 3. Liver enzyme and T-maze test. A. The serum level of liver enzyme in alcohol group mice. B. Change of T-maze test in alcohol mice. C. T-MAZE before alcohol binge. D. T-MAZE after alcohol binge. The data are presented as the mean ± standard error of the mean (SEM) and were statistically analyzed using a one-way ANOVA. * p < 0.05 difference in the group, **p < 0.05 difference compared with 11 days, 25 days, and 44 days # p < 0.05 difference compared with alcohol group

Round 2
Reviewer 1 Report
The authors improved their manuscript and It could be considered for publication
Author Response
We appreciate the Reviewer’s thoughtful comment.
Reviewer 2 Report
“Response 1: We appreciate the Reviewer’s thoughtful comment. When we designed our study, the pathology was paid to the company and commissioned. Regarding about neurology specialist, neurological examination was also performed with payment of research fee. All reports and results So, we did not mention about pathological specialist.”
Comment: Please add the name of the institute where the neurology and pathology tests were done in the text and state that this was a paid service.
Response 2: We apologize for causing confusions. We eliminated “maximum time 90s” and “8 weeks” replaced it with “maximum time 60s” and “6 weeks””
Comment: line 144, it is still 90 sec
“Response 3 and 4: We appreciate the Reviewer’s thoughtful comment. We thought that alcohol consumption had an effect instantly on cognitive function. Therefore, 1 day after, we measured T-maze test one more for normal state. As the reviewer mentioned that the sentence "For cognitive function, we performed T-maze studies weekly before and after alcohol binging" can cause confusion by measuring 2 times compared to fig 3B. So, we revised and added the following clarification in the "Experimental Animals" section: "For cognitive function, we performed T-maze studies 3 times before alcohol binge, after alcohol binge and 1 day after. Also, alcohol binge was provided at 5 g/kg/per 15 days, 40% ethanol." Additionally, we included nutrient sources and composition with table1.”
Comment: It is still not clear where the diets were obtained or purchased from. Please add the provider
Response 5: Thanks for raising this point. We trained only 1 time. We mentioned this at 2.3. Experimental Animals section.
“We trained mice once until the mouse found food.””
Comment: Is it enough to train the mice only once? Please discuss based on available literature and add this point to the discussion section
“Response 6, 7 and 14: In figure 1A and figure 3B, we clarified the T-maze performed date.
We gave chronic alcohol in diet group (HCD, HPD, HFD). Authors can see this fact in the Fig. 1A. We changed “10 days, 40 days” → “11 days, 44 days” And in figure 3B, we added the y-axis label.”
Comment: Figure 1A shows binging was done on days 11, 25, and 44, panels 3C and 3D however show data for 11, 25 and 40 days?!
Response 10: Thanks for these suggestions. We included the following sentence in the “Pathology” section: “Inflammation was classified from grades 0 to 3 (0: none, 1: 1–2 foci per ×20 field, 2: 2–4 foci per ×20 field, and 3: >4 foci per ×20 field).”
Comment: the statistical test is still missing
Response 11: As reviewer suggested, we extended to the high protein diet on Fig. 2.”
Comment: it is not extended
Comment: Lines 215-217” Interestingly, temporary recovery of cognitive function occurred during binging, but cognitive function resumed one day after binge” don’t you mean the decline in cognitive function resumed one day after?
Response 16: We agree with the reviewer’s comment and apologize for causing confusion regarding the among different panels of control and alcohol groups. In Figure 3B, we showed the average values for each day before binge drinking, after binge drinking and 1 day after binge drinking. Likewise, Alcohol group ~”
Comment: please provide a reason or if there was an error why did not you correct it in the revised version?
Response “As per the reviewer’s request, we included this sentence in the fig3 legend section: “The data are presented as the mean ± standard error of the mean (SEM) and were statistically analyzed using a one-way ANOVA(?).”
Comment: please state the posthoc test
Author Response
“Nutritional status and diet style affect cognitive function in alcoholic liver disease”
Point-to-point responses to comments by the Reviewer 2
First of all, we would like to thank the Reviewer2 for his/her comments, which helped us to improve this manuscript.
Specific Comments:
- Comment 1: Please add the name of the institute where the neurology and pathology tests were done in the text and state that this was a paid service.
- Response 1: We appreciate the Reviewer’s thoughtful comment. We added comments on pathology and Neuropsychological Test. “We commissioned a pathological report to the pathology department of a Chuncheon sa-cred heart hospital.” and “We commissioned the cognitive function test to the neurological function laboratory of Chuncheon Sacred Heart Hospital.”
- Comment 2: figure 1A, shows that animals were sacrificed 6 weeks after the study was initiated, is it 6 or 8 weeks? Also, was the max time in the maze 60 (figure 1B) or 90 sec (line 118)
- Response 2: We apologize for causing confusions. We eliminated “maximum time 90s” and “8 weeks” replaced it with “maximum time 60s” and “6 weeks”
- Comment 3: It is still not clear where the diets were obtained or purchased from. Please add the provider.
- Response 3: As reviewer recommended, we added the company name. “After clinical study analyses, we conducted alcohol animal experiments in mice fed a high-protein, high-carbohydrate, and high-fat diet purchased from Dooyeol Biotech (Seoul, Korea).”
- Comment 4:“We trained mice once until the mouse found food.” Is it enough to train the mice only once? Please discuss based on available literature and add this point to the discussion section.
- Response 4: We apologize for causing confusion. We trained mouse until the mice found food before starting the test. We changed sentence and we mentioned T-maze test on discussion section.
“We trained mouse until the mice found food before starting the test.”
“T-maze test has been used for the function with memory and spatial learning in an-imal model [31,32]. Different sizes and shapes of T- maze test might be used according to the types of animal model. In our animal model of alcohol-induced liver disease, one practice of reaching food was performed prior to the T maze test.”
- Comment 5: “Response 6, 7 and 14: In figure 1A and figure 3B, we clarified the T-maze performed date. We gave chronic alcohol in diet group (HCD, HPD, HFD). Authors can see this fact in the Fig. 1A. We changed “10 days, 40 days” → “11 days, 44 days” And in figure 3B, we added the y-axis label.”
Comment: Figure 1A shows binging was done on days 11, 25, and 44, panels 3C and 3D however show data for 11, 25 and 40 days?!
- Response 5: We made mistakes in Fig. We apologize for causing confusions. We changed all 40 to 44 days in Fig.
- Comment 6: Response 10: Thanks for these suggestions. We included the following sentence in the “Pathology” section: “Inflammation was classified from grades 0 to 3 (0: none, 1: 1–2 foci per ×20 field, 2: 2–4 foci per ×20 field, and 3: >4 foci per ×20 field).”
Comment: the statistical test is still missing
- Response 6: Thanks for rising this point. We add this sentence on Fig 2. “The data are presented as the mean ± standard error of the mean (SEM) and statistically analyzed using a Mann-Whitney test.”
- Comment 7: Response 11: As reviewer suggested, we extended to the high protein diet on Fig. 2.” Comment: it is not extended
- Response 7: Thanks for reviewer’s comment. We extended line of +alcohol of Fig 2. We can clearly see the extended line in Fig 2.
- Comment 8: Lines 215-217” Interestingly, temporary recovery of cognitive function occurred during binging, but cognitive function resumed one day after binge” don’t you mean the decline in cognitive function resumed one day after?
- Response 8: We apologize for causing confusions. Temporary recovery of cognitive function occurred AFTER binging NOT DURING binging. We revised the sentences.
Regarding resuming of cognitive function, we mean “the decline in cognitive function after continuous alcohol drinking resumed one day after” We changed sentence.
“Temporary improvement in cognitive function declined again the day after binge.”
- Comment 9: please provide a reason or if there was an error why did not you correct it in the revised version?
- Response 9: The reason of differences is that the groups of mice used are different in Fig . Fig3B and C, D. Fib 3B was performed with 20 mice to see the cognitive function of the alcohol model, and C and D used 9 mice in each group to see the changes in the alcohol model according to diet. We added methodology of Fig 3b at the method and Fig 3.
“We used 20 mice to evaluate changes of cognitive function in alcoholic liver disease. To induce the effect of diet in alcohol mice model, 45 mice were subdivided into 5 groups (n=9/group; normal regular liquid diet, control [5% EtOH liquid diet + regular liquid diet], high carbohydrate [5% EtOH + high-carbohydrate liquid diet (HCD)], high fat [5% EtOH + high-fat liquid diet (HFD)], and high protein [5% EtOH + high-protein liquid diet (HPD)]) and subject to the indicated conditions for 6 weeks.”
“A. The serum level of liver enzyme in alcohol group mice (n=20). B. Change of T-maze test in alcohol mice (n=20).”
- Comment 10: please state the posthoc test
- Response 9: As reviewer recommended, we stated the posthoc test in the Fig. 3. Regarding **p < 0.05 in Fig3B, increasing pattern is important in Anova test. We mentioned posthoc in the text.
“The data are presented as the mean ± standard error of the mean (SEM) and were statistically analyzed using a one-way ANOVA (LSD) and Mann-Whitney test. * p < 0.05 difference at the posthoc test (LSD); **p < 0.05 difference between the three groups (11 days, 25 days, and 44 days); # p < 0.05 difference compared with alcohol group.”
“In the posthoc analysis of alcohol group (11days, 25 days, and 44 days), all comparisons showed significant difference [mean times of escape latency (standard deviation) are 34 (8), 48 (5), and 58 (6) seconds in before alcohol binge, 21 (6), 26 (5), and 29 (4) in after alco-hol binge, and 36 (11), 42 (7), and 52 (5) in 1 day after binge, respectively].”